# Copper and Nickel Coating of Carbon Fiber for Thermally and Electrically Conductive Fiber Reinforced Composites

**DOI:** 10.3390/polym11050823

**Published:** 2019-05-07

**Authors:** Simon Bard, Florian Schönl, Martin Demleitner, Volker Altstädt

**Affiliations:** Department of Polymer Engineering, University of Bayreuth, Universitätsstr. 30, 95444 Bayreuth, Germany; florian.schoenl@uni-bayreuth.de (F.S.); martin.demleitner@uni-bayreuth.de (M.D.); altstaedt@uni-bayreuth.de (V.A.)

**Keywords:** Thermal conductivity, prepreg, carbon fiber, nickel coating, copper coating, flexural strength

## Abstract

In this paper, the thermal and electrical conductivity and mechanical properties of fiber reinforced composites produced from nickel- and copper-coated carbon fibers compared to uncoated fibers are presented. The carbon fibers were processed by our prepreg line and cured to laminates. In the fiber direction, the thermal conductivity doubled from ~3 W/mK for the uncoated fiber, to ~6 W/mK for the nickel, and increased six times to ~20 W/mK for the copper-coated fiber for a fiber volume content of ~50 vol %. Transverse to the fiber, the thermal conductivity increased from 0.6 W/mK (uncoated fiber) to 0.9 W/mK (nickel) and 2.9 W/mK (copper) at the same fiber content. In addition, the electrical conductivity could be enhanced to up to ~1500 S/m with the use of the nickel-coated fiber. We showed that the flexural strength and modulus were in the range of the uncoated fibers, which offers the possibility to use them for lightning strike protection, for heatsinks in electronics or other structural heat transfer elements.

## 1. Introduction

Efficient heat transport is important for increasing the lifetime and performance of parts [1,2]. Thermal conductivity reduces hotspots and disperses heat, which reduces stress concentration in the material. Especially polymeric materials, such as carbon fiber reinforced composites with epoxy matrices, inhibit low thermal conductivity and need efficient thermal management. Thermally conductive carbon fiber reinforced composites might be useful for heated rollers or the housing of lightweight electric engines. In contrast to metals, where electrons in the crystal structure can efficiently transfer the heat, phonons are the main transport mechanism for the heat transfer in amorphous materials as polymers [3,4,5]. Therefore, polymers exhibit only low thermal conductivities between 0.1 and 0.5 W/mK [6,7].

One possibility to increase the thermal conductivity is the incorporation of fillers, which is extensively studied with metallic [8,9,10], carbon-based [11,12,13,14] and ceramic materials [15,16,17]. Filler particles dispersed in epoxy resin lead to a number of matrix-particle interfaces. The phonon scattering at these interfaces can lower the thermal conductivity and mechanical properties of the materials. Additionally, the enhanced viscosity reduces processability and increases the cost.

Another pathway to increase the thermal conductivity is the use of pitch-based carbon fibers, also called graphite fibers. The morphology of these fibers fundamentally differs from those of the commonly used carbon fibers based on polyacrylonitrile (PAN) [18,19,20].

A further promising solution to enhance the conductivity is the coating of conventional carbon fibers with metals. The coating via galvanic methods is superior to the coating with Physical Vapour Deposition (PVD). In PVD processes, the thickness and distribution of the metal coating are difficult to control. Nickel is easier to process and, therefore, nickel coated fibers have been commercially available for several years (e.g., from Cycom or TohoTenax) [21]. Composites consisting of nickel coated fiber and epoxy resin were studied by Evans et al., who showed an enhancement in thermal conductivity in the fiber direction from 4.41 W/mK to 6.4 W/mK and in the transverse direction from 0.90 to 0.95 W/mK with 60 vol % fiber volume content [21]. Here, it is worth mentioning that the nickel coated fibers were only used in the outer layer of the laminate. In addition, composites of copper-coated fibers are known from the literature. Yu et al. used copper-coated fibers with very homogenous surfaces using rapid temperature annealing to produce fiber reinforced composites [22]. Copper is beneficial because of its thermal and electrical conductivity and the functional groups on the surface, which can bond to the matrix and presents fewer health concerns than nickel. The main challenge with copper is during galvanic processing, which leads to varying coating thicknesses. The thermal conductivity in this study was enhanced from 0.7 W/mK to 1.9 W/mK in the transverse and from 2.9 to 47.2 in the fiber direction. The fiber content was 48.5 vol % for the uncoated fiber and 26.2 vol % for the coated fiber.

Although the thermal conductivity of these composites was investigated partially, the mechanical properties and electrical conductivity remain uncertain. For use in industrial applications, an elaborative investigation about the thermal and electrical conductivity and the mechanical properties is necessary. This research, therefore, addresses the thermal conductivity in the transverse and fiber direction and the mechanical properties in bending tests.

## 2. Production and Characterization Methods

### 2.1. Materials

The resin tetraglycidylmethylenedianiline (TGMDA, Epikote^TM^ RESIN 496, Hexion Inc., Columbus, OH, USA) is four-functional with an epoxy equivalent of 115 g/eq and was cured with diethyltoluenediamine (XB3473^TM^, DETDA, hydrogen equivalent weight 43 g/eq).

The used carbon fibers were HTS40 (TohoTenax, Chiyoda, Japan), HTS40 with a nickel coating (TohoTenax, Chiyoda, Japan) and Grafil 34700 with a copper coating (Inca-Fiber, Chemnitz, Germany). The production of the copper-coated fiber is described in detail in the publications of Böttger-Hiller [23,24]. The mechanical properties can be found in Table 1. The thickness of the nickel-coating was 0.247 ± 0.01 µm and those of the copper-coating were 0.75 ± 0.2 µm. The diameter of the carbon fiber without coating was 7 µm for all fibers.

The volume share of nickel on the fiber was 12.6 %, and 33% of copper of the whole fiber.

### 2.2. Resin Preparation and Curing

The resin and hardener were stirred in a stoichiometric ratio of 72:28. The mixture had been degassed at 10–20 mbar. The samples were cured under pressure in a laboratory press at 120, 160 and 200 °C, at each temperature for 1 h with a heating rate of 10 K/min. A postcuring at 220 °C for 2 h followed, before cooling down at a rate of 5 K/min.

### 2.3. Prepreg Production and Consolidation

The unidirectional prepregs were produced via hot-melt processing using the laboratory scale prepreg impregnation machinery of the University of Bayreuth.

First of all, the unidirectional rovings of 12 K carbon fibers were pre-spread. A resin film was coated at 25 °C on a siliconized carrier paper in the coating unit of the prepreg machinery. Finally, the pre-spread fibers were impregnated with the resin film to the final prepreg with a calender (25 °C). The prepregs were then further processed via hand-layup. They were sealed in vacuum bags and cured under atmospheric pressure of 5 bar in a laboratory press at 120, 160 and 200 °C, at each temperature for 1 h with a heating rate of 10 K/min. A postcuring at 220 °C for 2 h followed before cooling down at a rate of 5 K/min.

### 2.4. Determination of Fiber Volume Content

Thermogravimetric measurements were conducted with the TG 209 F1 Libra (Netzsch-Gerätebau GmbH, Selb, Germany).

For the determination of the fiber volume content, a routine suggested by Monkiewitsch was followed, which was verified in a very thorough investigation [25]. In the suggested routine, fibers were dried for 2 h at 120 °C in the TGA. Then the fibers were heated in the TGA from 20 °C to 800 °C, with a heating rate of 2 K/min under a nitrogen flux of 85 ml/min. The samples of the laminates and samples from neat resin were also dried for 2 h in the TGA, then heated up to 450 °C with a heating ramp of 10 K/min. Finally, an isothermal step for 170 min at 450 °C was performed. All samples were handled with gloves to prevent possible contamination.

In the TGA, the fibers showed only a slight weight loss of (1.0 ± 0.1)%, which could be attributed to the oxidation of the sizing. During the drying step, no significant weight loss could be detected.

The fiber volume content ϕ could then be calculated:(1)$$\phi =\frac{m_{f}}{\rho_{f}\left(\frac{m_{f}}{\rho_{f}}+\frac{1-m_{r}}{\rho_{r}}\right)}$$
where $m_{f}$ is the mass of the fibers, $\rho_{f}$ is their density, and $\rho_{r}$ represents the density of the resin. The mass of the fibers was calculated by: (2)$$m_{f}=\frac{m_{l}-m_{r}}{1-m_{r}}$$
where $m_{l}$ is the remaining mass of the laminate after the cycle, and $m_{r}$ is the remaining mass of the resin.

The method was successfully verified and tested in comparison to the determination of the fiber volume content via density measurements [26].

### 2.5. Morphological Characterization

All samples were scanned with the Skyscan 1072 Micro-CT (Bruker, Artselaar, Belgium), with a linear resolution of 3.50 µm at a magnification of 80, with an accelerating voltage of 80 kV and tube current of 122 µA. The projection images were acquired over 180° at angular increments of 0.23° with an exposure time of 2.57 seconds per frame, averaged over six frames. Three-dimensional images were reconstructed using the reconstruction software provided by the manufacturer (NRecon Version 1.6.4.1), where the ring artifact reduction was applied as needed.

The samples were sputtered using the Cressington 108 Auto Sputter Coater with an Au coating thickness of 13 nm and studied in the JSM 6510 Scanning Electron Microscope (JEOL, Freising, Germany).

### 2.6. Thermal and Electrical Conductivity Measurements

The thermal conductivity was measured by the laser flash method (LFA) with LFA447 (Netzsch GmbH, Selb, Germany). Five shots were used with a duration of 30 ms each, the signal was fitted with the Proteus Analysis Software (Netzsch GmbH, Selb, Germany) by the Cape–Lehman algorithm. The tested samples had a diameter of 12.7 mm.

The electrical conductivity was measured in accordance with ASTM D257. Thereby, the electrical resistivity was measured with the Keithley 6517A for a resistivity higher than 100 MΩ, and for lower values, a Keithley 2100 (Keithley, Cleveland, OH, USA) with samples of 60 mm was used.

The density was measured with the AG245 (Mettler-Toledo International Inc., Columbus, OH, USA) using Archimedes’ principle. The thermal heat capacity was measured with the DSC 1 (Mettler-Toledo International Inc., Columbus, OH, USA), according to ASTM E1269–11, with a heating rate of 20 K/min.

### 2.7. Flexural Modulus and Strength

The mechanical properties of the composites were measured using the Zwick Z2.5. The samples for testing were prepared according to DIN EN ISO 14125 standards. The bending strength was measured on at least five samples of length 100 mm, width 15 mm and thickness 2 mm, by a three-point bending technique at a cross-head speed of 1 mm/min.

## 3. Results and Discussions

### 3.1. Morphology of Metal Coated Fibers and Laminates

The fibers were used as delivered from the supplier. Figure 1 shows the cross-section records of the uncoated carbon fiber. The surface of these fibers was quite smooth, only the cut edge of the fiber showed detached layers. Figure 2 shows a very homogenous nickel coating on the surface of the carbon fibers. The adhesion between the nickel and fiber was excellent. Figure 3 shows the surface of the copper-coated fiber. The coating of copper was much less homogenous: Larger agglomerations could be found on the surface. In addition, the adhesion with the fiber was not as smooth as those of the nickel fibers: Parts of copper seemed to detach from the fiber. Already during production, smaller particles loosened from the fibers, however, their share was determined to be below 2%.

Figure 4 shows the cross-section of the laminates in SEM. The sample from the copper-coated fiber laminate showed an heterogeneous distribution of the copper coating. Therefore, the coating thickness on the fibers varied. The nickel coating was much more homogenous.

The µCT records in Figure 5 confirmed the excellent laminate quality. The nickel-coated laminate was very homogenous. The pictures on the left show the void content. Here, the nickel-coated laminate showed no visible voids, the software calculated them as below 0.5%. The copper samples showed some minor voids, which were below 0.8%. The image in Figure 5d shows some heterogeneity of the copper-coated sample.

### 3.2. Thermal Conductivity of Laminates

The thermal conductivity can be calculated from the heat capacity $c_{p}$, density ρ and thermal diffusivity a by the following equation:(3)$$\lambda =\rho *c_{p}*a$$

The densities at room temperature are shown in Table 2. The heat capacity was determined to be 1.11 ± 0.040 J/gK for the resin hardener system, 0.966 ± 0.045 J/gK for the PAN fiber at room temperature, 0.864 ± 0.039 J/gK for the nickel-coated fiber and 0.870 ± 0.050 J/gK for the copper-coated fiber. The heat capacity of the resin was in accordance with the findings of Baller [27]. Rana et al. determined the heat capacity of the carbon fiber to be 0.92 J/gK, so the measured value seems valid [28]. Knowing these values, the heat capacity of the samples can be calculated. The density as measured by Archimedes’ principle, and the diffusivity as measured by the laser flash method, can be found in Table 2.

The samples are denoted by their fiber volume content, which was determined by thermogravimetry (as described in the methodology section). Therefore, PAN_39.4 refers to a laminate produced from an uncoated PAN-based carbon fiber with a fiber content of 39.4 vol %. By comparing the density measured by the Archimedes’ principle and the density calculated from the contents derived from TGA measurements, the void content of the samples was determined. All samples showed void contents below 1%.

The thermal conductivity of the samples was measured in the transverse and in-fiber direction. The conductivities in the fiber direction are shown in Figure 6. In the fiber-direction, the thermal conductivity is expected to follow the rule of mixture and, therefore, a linear correlation was expected. The conductivities of the laminates from the uncoated fiber follow the equation 0.23 + 0.054 × x, where x is the fiber volume content of the PAN fiber. By this, the thermal conductivity of the PAN fiber in the fiber direction was calculated as 5.63 W/mK. For the nickel-coated fiber, the correlation from three measurement points was 0.23 + 0.102 × x, so the conductivity of the fiber should be close to 10.43 W/mK. From the two data points of the copper-coated fiber laminates, the thermal conductivity followed the correlation 0.23 + 0.385 × x, so the thermal conductivity of the copper-coated fiber was estimated as ~39 W/mK. The thermal conductivity of neat nickel was 91 W/mK and 391 W/mK for copper.

The thermal conductivities transverse to the fiber direction can be found in Figure 7. Unlike the thermal conductivity in the fiber direction, the transverse conductivity did not follow a linear trend. According to the models of Nielsen, Hatta/Taya or Tsai, the conductivity in filled polymer systems follows an exponential trend [29,30,31]. This trend seemed clearly visible for the carbon fibers.

The carbon fibers were radially isotropic and consisted of several graphitic layers, which were wrapped around the center [32]. Along the crystal structure of carbon, the thermal heat can be transferred efficiently. Between the different layers of the fiber, act van der Waals forces, which lead to low thermal transport [33]. In conclusion, mainly the outer shells of the carbon fibers were responsible for the thermal conductivity transverse and in-fiber direction. This explains the relatively low thermal conductivity of PAN-based carbon fibers compared to CNT, graphene and other carbon materials. Dong et al. determined the thermal conductivity of carbon fiber to be 10.2 W/mK in the fiber direction and 1.256 W/mK transverse to the fiber [34]. Rolfes et al. estimated the conductivities to ~7 W/mK and ~2 W/mK in the fiber and transverse direction [32]. The laminate from the copper-coated sample showed extraordinary high thermal conductivity compared to the other samples. The distances between the fibers were very small and the fibers were in close contact. Therefore, the number of copper-matrix interfaces decreased, which led to phonon scattering and thereby reduced the thermal conductivity.

The experimental values were then compared to models of Lewis and Nielsen. This model is used in various publications to calculate the thermal conductivity of composites [32,35,36,37]. In elaborative literature reviews, Progelhof et al. [38] and Pal [39] concluded that the Lewis–Nielsen model showed the most reliable results for compounds of rigid bodies. The conductivity of the composite is determined by:(4)$$\lambda_{k}=\lambda_{M}\frac{\;1+A\;B\;\phi }{1-B\;\phi \;C}$$
while ϕ is defined as the filler content and $\lambda_{M}$ as the conductivity of the matrix. The factors A, B and C reflect the filler geometry, orientation and thermal conductivity. According to Guth [40], the parameter A can be calculated with the aspect ratio p by: (5)$$A=\frac{p}{\left[2\ln\left(2p\right)\right]-3}+1$$

*B* is not an independent variable as it also reflects conductivity of filler and matrix and *C* reflects the maximal packing density $\phi_{max}$:(6)$$B=\frac{\frac{\left(\lambda_{F}\right)}{\left(\lambda_{M}\right)}-1}{\frac{\left(\lambda_{F}\right)}{\left(\lambda_{M}\right)}+A}$$
(7)$$C=1+\frac{\left(1-\phi_{max}\right)}{\phi_{max}^{2}}\phi$$

The maximum packing density of fibers in the composite was 82 % [41]. B is calculated with the thermal conductivity of the matrix given above and the transverse thermal conductivity of the carbon fiber of 2 W/mK. The transverse thermal conductivity of the fiber can only be estimated, as no method to measure a single fiber could be identified in the literature. Rolfes and Hammerschmidt [32] calculated a transverse conductivity of 2 W/mK from their experimental data for a round-type PAN fiber. For the metal-coated fiber, the through-thickness conductivity was calculated by the volume fraction of the metal and carbon fiber multiplied with the thermal conductivity of the metal, which was 91 W/mK for nickel and 391 W/mK for copper. A was calculated to 0.83 with an aspect ratio of 0.5, and Rolfes and Hammerschmidt confirm that this value was suggested in the literature [32]. The theoretical and experimental values can be found in Figure 8.

As it can be seen from the figure, the model was not applicable to the copper-coated and nickel-coated laminates. The model significantly underestimates the thermal conductivity of these laminates. In addition, the shape of the copper-coating was not comparable to the shape of typical fillers, of which the models are tailored to. The shell geometry of the metal coating around the fibers most likely transports the heat more efficiently than expected and thereby leads to higher thermal conductivities.

A main advantage of the copper-coated fibers compared to particles is that the structure resulting in the laminates is beneficial for the transport of heat. The more interfaces between particles and matrix are generated, the higher the thermal resistance due to phonon scattering [6]. In the laminates from coated fibers, the fibers are likely to bump into each other. The resistance between the metal layer of adjacent fibers is much smaller than those between copper and epoxy matrix. [22] This effect then leads to the enhanced thermal conductivities in the laminates with coated fiber.

### 3.3. Electric Conductivity

In polymers, the dominating mechanism for the transport of heat is phonon transport. In metals and carbon allotropes, the heat is transported mainly by electrons. This mechanism is much more efficient due to the much smaller radius of 10^−5^ nm of electrons compared to the free wavelength of phonons of 0.7 nm. [5,42,43] For this reason, the transport mechanism of the electric conductivity and the thermal conductivity is similar in carbon-based materials and metals. An increase in thermal conductivity of carbon or metal can be correlated with an increase in electrical conductivity.

Figure 9 shows the electric conductivities of the laminates. The laminates from the copper-coated fibers showed the highest conductivities compared to the nickel-coated and uncoated fibers at the same fiber content. It is was clear that the metal-coated fibers showed electric conductivities, which were three orders of magnitude higher than their uncoated counterparts. For the movement of electrons in heterogeneous materials, the distance between the particles is crucial. Once a conductive network of adjacent particles is built, the conductivity significantly increases. The filler content, which marks a sudden increase in electric conductivity is called the percolation threshold. The morphology of the coated laminates is beneficial to build a conductive network: The copper coating around the fibers easily makes contact with the surrounding copper shells. By this way, a conductive path out of the copper coating is formed.

### 3.4. Mechanical Properties

The flexural strength of the laminates can be found in Figure 10. For the uncoated laminate, the flexural strength increased with increasing fiber volume content up to 2068 ± 70 MPa with a fiber volume content of 60 vol %. The flexural strength decreased at a higher fiber volume content because of the higher number of fibers in direct contact without a matrix in between. The missing adhesion between the fibers then lowered the mechanical properties of the composite.

The flexural strength of the copper-coated fiber at 39.5 vol % was significantly lower than its uncoated counterpart at 39.4 vol % of carbon fiber. At a 48.1% and 49.4% fiber volume content, nickel-coated fibers showed a flexural strength comparable to the uncoated laminates. It seemed that the functional groups of the metal coated fibers led to an excellent adhesion between the fiber and matrix, which led to flexural properties in the range of the laminates from the uncoated fiber. In addition, the flexural modulus of the laminates from the coated and uncoated fibers was comparable. Only at very high fiber volume contents of 55.2 vol %, the nickel-coated fiber showed significantly lower mechanical properties than those of the uncoated PAN-based fiber. The microscopy records and micro-CT scans gave no rise to the assumption that the laminate quality was responsible for the lower mechanical properties. Still, the hand layup may have led to imperfections and should be responsible for the lower mechanical properties. The copper-coated fibers showed lower flexural strength and modulus than the other fibers. Here, the heterogeneous coating and the thickness of the coating might lead to a detachment of fiber and matrix and, therefore, weaken their properties. The effect of the replacement of the carbon fiber by a copper coating might be even stronger: Copper shows significantly lower flexural strength of 300–500 MPa (depending on specific composition) than carbon fiber reinforced laminates.

The flexural strength of laminates from metal-coated fibers was not determined in the literature so far. Only Evans et al. showed that the compressive strength of laminates from nickel-coated fibers was significantly lower than those of the laminates with an uncoated fiber [21]. Compared to aluminum, which shows higher densities (2.7 g/cm³), the flexural strength of the laminates in the underlying research was at a much higher level (aluminum 6061-T6: 310 MPa). In addition, the flexural modulus was higher compared to its metal counterpart (aluminum 6061-T6: 69 GPa).

## 4. Conclusions

The aim of this work was to explore the influence of a metal coating on the thermal and electrical conductivity and mechanical properties of laminates. We could show that the metal coating was very effective to enhance the thermal conductivity, both in the transverse and in-fiber direction. For the in-fiber direction, the thermal conductivity doubled from ~3 W/mK for the uncoated fiber to ~6 W/mK for the nickel, and increased six times to ~20 W/mK for the copper-coated fiber for a fiber volume content of ~50 vol %. The thermal conductivity transverse to the fiber was 45% higher with the nickel coating and enhanced by 380% in the copper coating of the carbon fibers, compared to the uncoated fibers. A linear trend for the correlation between fiber volume content and thermal conductivity in the fiber direction was found for all composites. Perpendicular to the fibers, the thermal conductivity followed the equations suggested by Lewis and Nielsen for the uncoated fiber laminates. The electric conductivity was three orders of magnitude higher than those of the laminate from uncoated PAN fiber. The mechanical properties were very promising and the flexural strength as the modulus was in the range of the uncoated fiber laminates.

## Figures and Tables

**Figure 1 polymers-11-00823-f001:**
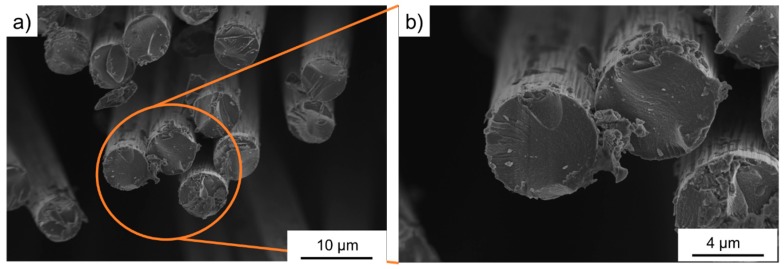
SEM cross-section images of the uncoated carbon fiber (**a**) overview picture, (**b**) marked section magnified.

**Figure 2 polymers-11-00823-f002:**
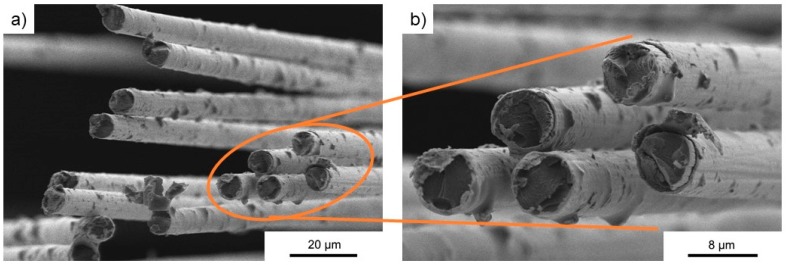
SEM cross-section images of the nickel-coated fiber (**a**) overview picture, (**b**) marked section magnified.

**Figure 3 polymers-11-00823-f003:**
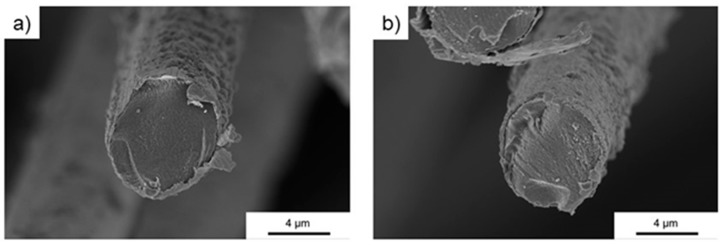
SEM cross-section images of the copper-coated fiber (**a**) with inhomogenous coating, (**b**) with detached coating.

**Figure 4 polymers-11-00823-f004:**
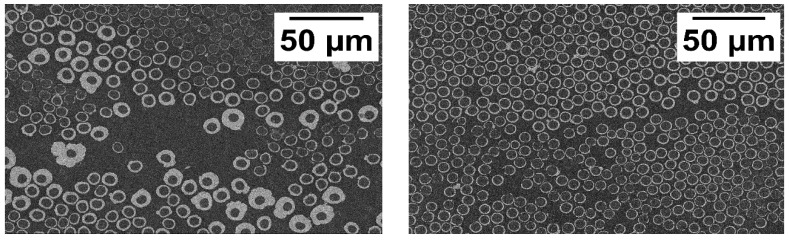
SEM cross-section images of the laminates from (**left**) the copper-coated fiber (39.5 vol % fiber) and (**right**) nickel-coated fiber (49.4 vol % fiber).

**Figure 5 polymers-11-00823-f005:**
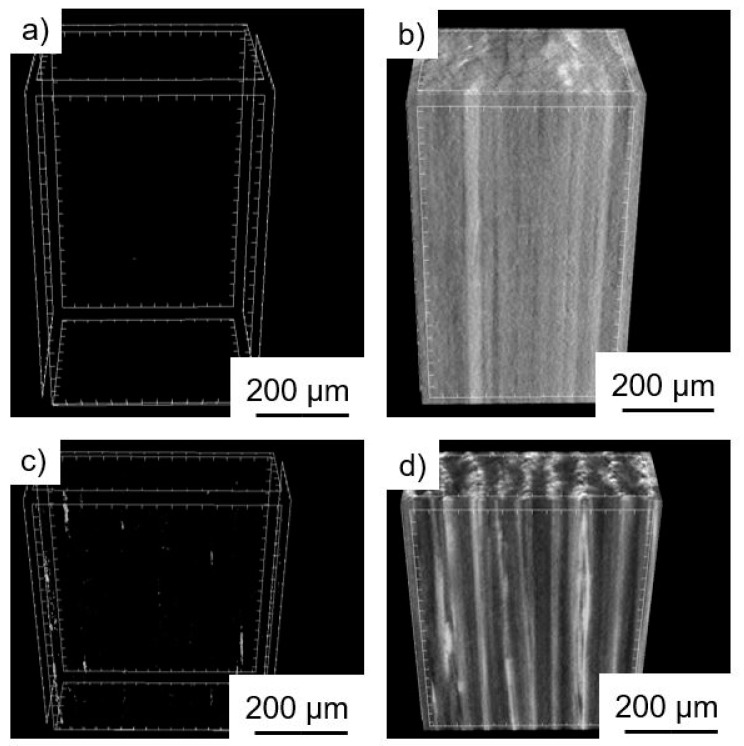
Micro-CT scans from the nickel-coated sample with 55.2 vol % fiber (**a**,**b**) and copper-coated sample with 39.5 vol % fiber (**c**,**d**). The images in (**a**,**c**) show the voids of the laminates.

**Figure 6 polymers-11-00823-f006:**
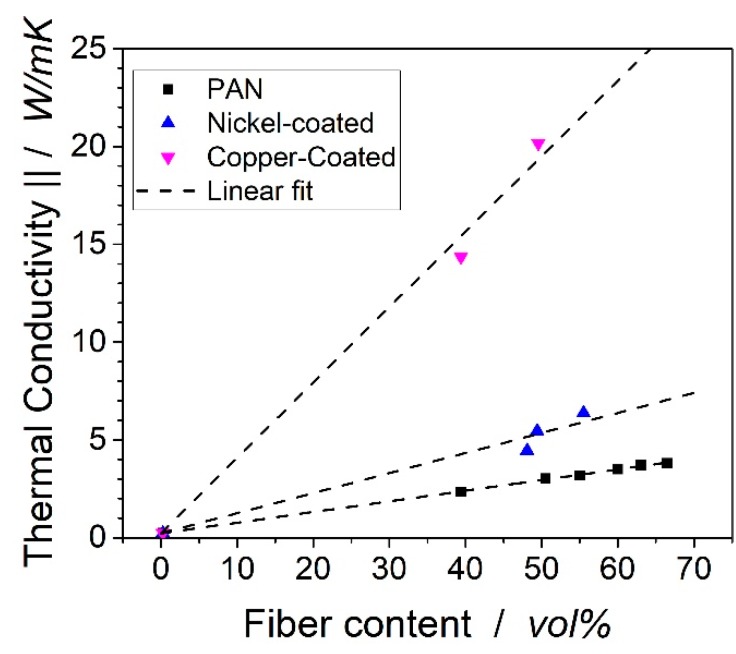
Thermal conductivity vs. fiber volume content for the in-fiber direction. The standard deviation is not visible in the graphs.

**Figure 7 polymers-11-00823-f007:**
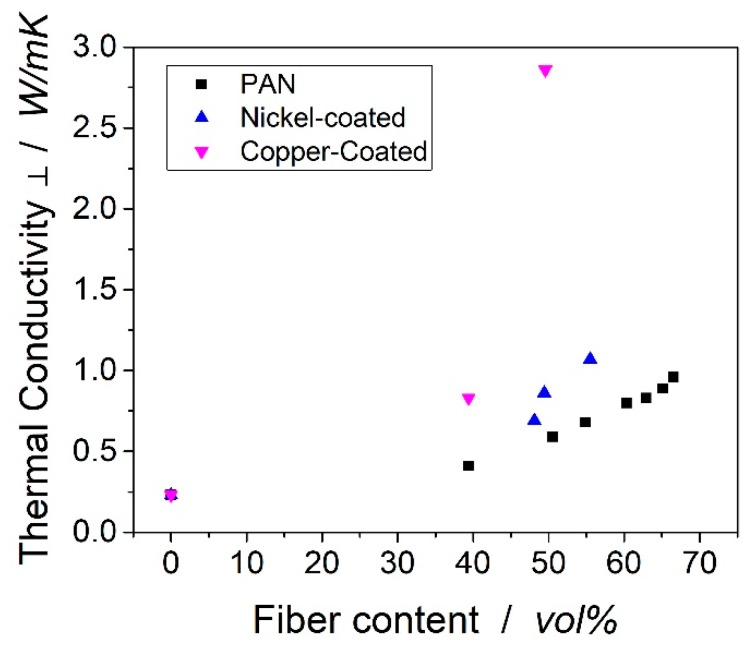
Transverse thermal conductivity of laminates vs. fiber volume content. The standard deviation is not visible in the graphs.

**Figure 8 polymers-11-00823-f008:**
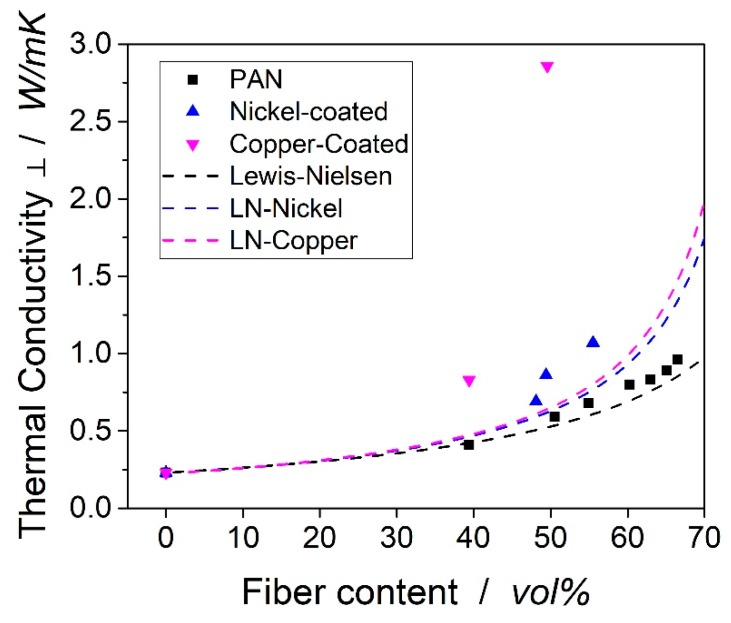
Experimental data versus analytical data from the presented model.

**Figure 9 polymers-11-00823-f009:**
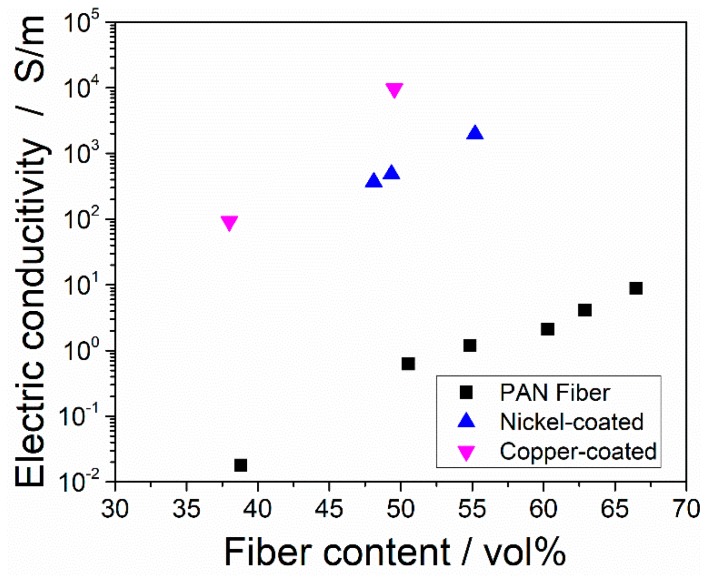
Electric conductivity of uncoated and metal-coated laminates.

**Figure 10 polymers-11-00823-f010:**
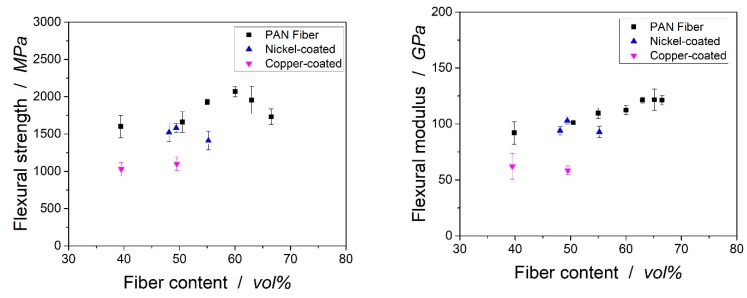
Flexural strength (**left**) and modulus (**right**) of uncoated and metal-coated laminates.

**Table 1 polymers-11-00823-t001:** Properties of fibers used in the underlying research.

Fiber	Manufacturer	Tensile Strength	Tensile Modulus
-	-	MPa	GPa
HTS40	TohoTenax	4620	239
HTS40_NC (Nickel coated)	TohoTenax	2900	230
Grafil_34700 (Copper-coated)	Inca-Fiber GmbH	n.a.	n.a.

**Table 2 polymers-11-00823-t002:** The heat capacity, density and thermal diffusivity of the tested samples at 20 °C, the sample name consists of the fiber type and the fiber volume content.

Sample	Heat Capacity	Density	Diffusivity ⊥	Diffusivity ||
-	J/gK	g/cm³	m²/s	m²/s
PAN_39.4	1.053	1.39 ± 0.010	0.29 ± 0.005	1.60 ± 0.005
PAN_50.5	1.037	1.45 ± 0.012	0.39 ± 0.003	2.01 ± 0.004
PAN_54.9	1.030	1.47 ± 0.009	0.45 ± 0.004	2.18 ± 0.006
PAN_60.3	1.023	1.50 ± 0.014	0.51 ± 0.007	2.28 ± 0.007
PAN_62.9	1.019	1.51 ± 0.008	0.54 ± 0.005	2.40 ± 0.003
PAN_66.5	1.014	1.53 ± 0.011	0.61 ± 0.008	2.45 ± 0.004
NC_48.1	0.992	1.90 ± 0.015	0.37 ± 0.003	2.36 ± 0.003
NC_49.4	0.989	1.92 ± 0.090	0.45 ± 0.002	2.87 ± 0.005
NC_55.2	0.974	2.01 ± 0.007	0.55 ± 0.003	3.26 ± 0.003
CC_39.5	1.019	2.33 ± 0.012	0.35 ± 0.005	6.04 ± 0.004
CC_49.5	0.991	2.67 ± 0.007	1.075 ± 0.003	7.62 ± 0.006

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
