# Peer review of "Copper and Nickel Coating of Carbon Fiber for Thermally and Electrically Conductive Fiber Reinforced Composites"

_polymers, 2019, doi:10.3390/polym11050823_

Round 1
Reviewer 1 Report
The manuscript reports the mechanical, thermal and electrical properties of composites reinforced with copper and nickel coated carbon fibers. The topic is of interest in the field. However, I would like to have below issue addressed before publication.
1. The application of carbon fiber reinforced composites including structural and non-structural. The authors should provide more clear application target in introduction. As properties such as thermal and electrical conductivity are not crucial for structural applications. For example, specific examples of industrial application could be provided.
2. For mechanical properties, the authors studied the flexural strength. But a concern in this study that has not been commented is the adhesion between copper coated fiber and resins. I would like to see tensile properties provided here, as good adhesion between fiber/matrix could enhance the overall composite properties. Otherwise, the composite will tend to fail. It is another reason a clear application direction is needed.
3. What is the nominal diameter of the coated carbon fibers?
4. The term “prepreg” presents several times in the manuscript, but it seems that most of the time it means the cured composite. Please check and fix.
5. Page 3, line 100. Please provide some brief discussion on the volume content determination procedure.
Author Response
Dear Reviewer, please find the response to your comments in the attached Cover-Letter.

Reviewer 2 Report
The paper entitled “Copper and nickel coating of carbon fiber for thermally and electrically conductive fiber reinforced composites” describes processing and the characterization of carbon fibers coated with nickel and copper. The thermal and electrical conductivity and mechanical properties of fiber reinforced composites produced from nickel- and copper coated carbon fibers compared to uncoated fibers are investigated. The carbon fibers were processed by our prepreg line and cured to laminates. In fiber direction, the thermal conductivity was doubled from ~3 W/mK for the uncoated fiber to ~6 W/mK for the Nickel and increased six times to ~20 W/mK for the copper-coated fiber for a fiber volume content of ~50 vol%. Transverse to the fiber, the thermal conductivity increased from 0.6 W/mK (uncoated fiber) to 0.9 W/mK (nickel) and 2.9 W/mK (copper) at the same fiber content. Also the electrical conductivity could be enhanced to up to ~1500 S/m with the use of the nickel-coated fiber. The flexural strength and modulus are in the range of the uncoated fibers, which offers the possibility to use them for lightning strike protection, for heat sinks in electronics or other structural heat transfer elements.
The paper is so quite interesting and different characterizations are reported in this work to investigate and analyse the effect of coating treatment of carbon fibers.
The manuscript will be a useful contribution in Polymers after minor revision. Some modifications may be brought to improve the overall quality of the manuscript.
I have no hesitation to suggest the publication.
Specific Comments:
Morphology of metal coated fibers and prepreg laminates-Paragraph 3.1-Figure 1 and 2: The authors are invited to introduce the morphological investigation of uncoated fibers to evaluate better the effect of coatings on the fiber surfaces. The authors are invited also to modify the manuscript introducing the comment relative to the uncoated fibers.
Author Response
Dear Reviewer, thank you very much for the comments. The changes were made accordingly. Please find the response to your comments in the attached.

Round 2
Reviewer 1 Report
The authors have addressed the comments provided previously. Recommend for publication.